# Candidate Transcript Panel in Semen Extracellular Vesicles Can Improve Prediction of Aggressiveness of Prostate Cancer

**DOI:** 10.3390/ijms26199562

**Published:** 2025-09-30

**Authors:** Adriana Ferre-Giraldo, Manel Castells, Alicia Madurga, Ariadna Arbiol-Roca, Maurizio de Rocco-Ponce, Lluís Bassas, Francesc Vigués, Sara Larriba

**Affiliations:** 1Immune-Inflammatory Processes and Gene Therapeutics Group, Genes, Disease and Therapy Program, Bellvitge Biomedical Research Institute (IDIBELL), 08908 Hospitalet de Llobregat, Barcelona, Spain; aferre@idibell.cat; 2Urology Service, Bellvitge University Hospital-ICS, 08908 Hospitalet de Llobregat, Barcelona, Spain; mcastells@bellvitgehospital.cat (M.C.); fvigues@bellvitgehospital.cat (F.V.); 3Territorial Clinical Laboratory Metropolitan South, Bellvitge University Hospital, 08908 Hospitalet de Llobregat, Barcelona, Spain; amadurga@bellvitgehospital.cat (A.M.); ariadna.arbiol@bellvitgehospital.cat (A.A.-R.); 4Andrology Service-Puigvert Foundation, Instituto de Investigaciones Biomédicas Sant Pau (IIB-Sant Pau), 08025 Barcelona, Barcelona, Spain; mderocco@fundacio-puigvert.es (M.d.R.-P.); lbassas@fundacio-puigvert.es (L.B.)

**Keywords:** prostate cancer, mRNA, semen extracellular vesicles, biomarker, non-invasive diagnosis/prognosis

## Abstract

The need for prostate cancer (PCa)-specific biomarkers that enable more accurate detection of the disease and better prediction of tumor aggressiveness remains ongoing due to the low cancer specificity of PSA screening. Several potential mRNA markers for diagnosing PCa, in tissue and urine, have been reported in the literature. In this study, we aim to explore the potential of selected prostate-specific molecules and transcripts contained in small extracellular vesicles (sEVs) in semen to predict PCa risk reclassification for patients with moderately elevated PSA levels—a clinical scenario where identifying truly non-invasive biomarkers is especially critical. RT-qPCR analysis in semen sEVs successfully showed differential expression of *KLK3* and *PCA3* genes between PCa and healthy controls, whereas *CREB3L4*, *CCNQ* and *DUSP23* levels were related to the severity or degree of PCa affectation. Our findings also present strong evidence that classifiers based on combined long transcript levels in semen sEVs serve as effective biomarkers. They can be used alone or in combination with blood PSA and/or semen citric acid levels to improve the diagnosis of PCa and assess its severity and disease progression with high accuracy. This strategy would allow a more comprehensive assessment, increase prognostic accuracy, and facilitate accurate clinical decision-making in the management of PCa.

## 1. Introduction

Prostate cancer (PCa) screening has extended worldwide with the aim of detecting clinically significant cancer at an early stage when treatment could be more effective, leading to a reduction in cancer-related deaths. PCa screening mainly involves the determination of the levels of plasmatic prostate-specific antigen (PSA) combined with a physical examination of the prostate gland (by digital rectal exam—DRE) and/or a multiparametric magnetic resonance imaging (mpMRI) assessment revealing patients with potential malignancy. Diagnosis should then be confirmed in a transrectal or transperineal biopsy, which can further evidence the severity or degree of affectation by means of the modified Gleason score (GS) [1].

Although specific for prostatic tissue, the limitations of blood PSA as a biomarker are well established [2,3], attributable to its recognized low cancer specificity. The specificity is low because, in addition to prostate cancer, a number of benign conditions such as benign prostatic hyperplasia (BPH) and prostatitis can cause elevated PSA levels. This particularity leads to the generation of false positive results with the consequence of over-diagnosis and unnecessary biopsies of benign disease (only in about 30–40% of biopsied men is the PCa diagnosis confirmed: the chance of having PCa is over 50% if the PSA level is over 10 ng/mL whereas it is only about 20% in men with a PSA level in the range from 4 to 10 ng/mL, thus requiring special attention). This fact highlights the necessity of identifying new fluid biomarkers to eventually complement PSA and increase specificity. Additionally, PSA levels do not correlate with tumor aggressiveness, which could result in overtreatment of indolent tumors. Specifically, non-invasive prostate cancer-specific biomarkers that can distinguish between the aggressive prostate cancer tumor type and the indolent prostate cancer form are urgently needed to increase PCa screening accuracy and help clinicians provide personalized treatment.

Some efforts have been made to detect prostate-specific proteins, RNA, or other molecules in urine for their use as biomarkers to detect and monitor PCa. Common PCa urine biomarker tests include *PCA3* (prostate cancer antigen 3) RNA. *PCA3* is a prostate-specific long non-coding RNA (lncRNA) which is highly overexpressed in PCa cells relative to adjacent benign cells [4]. It can be detected with the commercially available Progensa PCA3 assay, which utilizes the whole urine collected after prostate massage (DRE). The DRE helps release prostate cells from the posterior part of the gland, including those containing *PCA3*, into the urethra, making them detectable in the urine sample. This test has improved the diagnosis of PCa, but its diagnostic value for aggressive PCa is limited. *PCA3* is also included in the ExoDx Prostate test, a non-invasive urine assay that does not require DRE and analyzes the gene expression of three genomic markers contained in exosomes (a small-sized subtype of extracellular vesicles, EVs) that are elevated in high-grade prostate cancer: *PCA3*, *ERG* (V-ets erythroblastosis virus E26 oncogene homologs) and *SPDEF* (SAM-pointed domain-containing Ets transcription factor) [5].

Semen is emerging as a likely source of PCa biomarkers since the primary function of the prostate gland is to produce seminal fluid, which is a crucial component of semen. Prostatic secretions, derived from all parts of the prostate when prostate muscle contracts, account for 30–40% of semen volume, and contain proteins, organic acids, and other molecules involved in sperm motility and viability for reproduction. Changes in prostatic secretions can be indicative of prostate-affecting diseases such as inflammation and cancer. Among prostatic proteins, PSA is secreted by normal and malignant epithelial cells in the prostate. This protein, encoded by the *KLK3* gene, is a serine protease that regulates the fluidity of semen [6]. The *KLK3* gene is positively regulated by the androgen signaling pathway. Another important molecule secreted by the prostate is citrate, the final secretory organic acid required for the mitochondrial tricarboxylic acid (TCA) cycle. Citrate metabolism is drastically reprogrammed during prostate carcinogenesis [7], as cancer cells utilize citrate to produce fatty acids, thus resulting in the depletion of citrate in tumors but not in the case of BPH. Some groups have suggested that intratumoral citrate as well as that found in semen or in urine shows higher cancer specificity and may serve as a more reliable biomarker than PSA in blood [8,9,10].

Interestingly, seminal plasma, the fluid component of semen, contains a high concentration of EVs that originate along the male reproductive tract, such as the prostate. These vesicles contain DNA, RNA, and proteins that can reflect the health of the cell of origin. Analyzing prostate gene expression in the small EVs (sEVs) from semen could be an attractive non-invasive procedure for PCa diagnosis, as we previously demonstrated for semen sEV sncRNAs [11,12].

Within this context, the need for prostate cancer-specific biomarkers that allow for more accurate detection of PCa and improve the prediction of tumor aggressiveness continues. New putative mRNA markers for PCa diagnosis in tissue and/or fluids such as urine have been described in the literature. In this study, we chose a targeted approach to investigate the capacity of selected prostate-specific molecules and sEV-contained transcripts, in semen as fluid, to predict PCa risk reclassification in patients with moderately elevated PSA levels, where the identification of truly non-invasive biomarkers for PCa is more necessary.

## 2. Results

### 2.1. Selection of Candidate RNA Transcripts to Study in Semen sEVs

To select candidate genes for validation as PCa biomarkers in sEVs from semen samples, we relied first on a preprint scientific article (https://doi.org/10.1101/2024.08.21.608965) that identified biomarkers that help to discern the GS in prostate cancer patients. This study proposed gene expression-based biomarkers for PCa by integrating microarray expression datasets, including more than 1.5 million samples, and applying machine learning models, considering annotations standardized to GS classifications. The proposed biomarkers achieve a minimum precision of 0.80, and their ROC-AUC profiles (≥0.70) align with those of other recently proposed innovative biomarkers, making them promising candidates for further investigation. Among the proposed candidate biomarkers, we selected *CREB3L4* and its family member *CREB3L2* [also involved in prostate cancer signaling (KEGG)], *CCNQ*, *DUSP23*, *ACTG1*, and *MXD4,* since they are described to be preferentially expressed in the prostate among the organs of the reproductive tract.

Additionally, other genes were selected as they are tested in urine EV samples for PCa diagnosis, such as *PCA3* and *KLK3* (the gene that codes for the PSA protein in the prostate).

### 2.2. Clinical Evaluation of Individuals Included in the Study

The clinicopathological features of patients and control individuals are shown in Table 1 and Appendix A. In terms of prostate-specific antigen (PSA) levels, most of the prostate cancer (PCa) cases and benign prostatic hyperplasia (BPH) controls were within the diagnostic “grey zone” of PSA values (4–10 ng/mL). Only four individuals presented with PSA levels that were slightly above this threshold (10.46–17 ng/mL). Both groups, BPH and PCa, presented similar age values (mean value: 58.8; *p*-value: 0.906).

The histopathological evaluation of the prostate biopsy showed that most patients (24 out of 27 individuals) showed mild disease, with PCa in early stages [GS6 and GS7(3+4)].

The classification of PCa samples into prognostic groups in accordance with the AJCC (American Joint Committee on Cancer) PCa staging system (8th edition) showed that a considerable number of samples (22 out of 27 samples) belonged to low/intermediate-risk tumors (stage I, IIA, and IIB), whereas 5 samples (18.5% of PCa samples) were classified as high-risk tumors (unfavourable intermediate IIC and high-risk–advanced IIIB).

### 2.3. PSA and Citric Acid in Semen Compared with PSA in Blood

PSA protein and citric acid in semen were also added as potential seminal biomarkers as both are secreted by normal prostate tissue and are linked to prostate health, representing potential complementary tests for PCa diagnosis.

The levels of PSA and citric acid were quantified in semen samples from HCt (healthy controls), BPH and PCa individuals (Table 1) and compared with PSA levels in blood. PSA and citric acid levels in semen were decreased in BPH and PCa compared to HCt controls, although the difference was statistically significant (*p*-value: 0.012) only for citric acid in the PCa vs. HCt comparison. Conversely, PSA in blood was differentially overexpressed in PCa and BPH when compared to HCt individuals (*p*-value < 0.0001). No difference in expression was observed between BPH and PCa groups for PSA or citric acid in semen, as it occurred for PSA in blood (Figure 1), and neither between GS6 and GS7 samples.

### 2.4. Altered Expression of mRNAs Contained in Semen sEVs

The presence of differences in sEV-contained transcripts was evaluated first among HCt, BPH, and PCa individuals. Only the *KLK3* gene was significantly underexpressed in BPH compared to HCt controls (*p*-value: 0.002), whereas *KLK3* (*p*-value: 0.006) and *PCA3* (*p*-value < 0.0001) were differentially expressed between PCa and HCt individuals (Figure 2; Appendix A).

When the PCa samples were classified by the histopathological pattern, differences in expression were found, specifically between HCt and PCa_GS6 for *KLK3* and *PCA3* as well as between HCt and PCa_GS7 for the *PCA3* gene, although no difference was found between GS6 and GS7 samples (Appendix A). Interestingly, when PCa samples were classified by the AJCC prognostic criteria, differences in expression between low- and high-risk PCa were found for *CCNQ* and *DUSP23* genes (Figure 3).

No gene was differentially expressed between the BPH and PCa groups; a similar occurrence was found with PSA in blood. Interestingly, no differences were found among vasectomized and non-vasectomized individuals for each group (Appendix A), corroborating the prostatic origin of these semen sEV-contained mRNAs.

In order to assess a potential relationship between these sEV transcript levels and age (as a confounding factor), a Spearman correlation analysis was performed. Considering all the samples in the study (HCt and BPH_PCa samples), the analysis revealed a significant correlation between age and molecular variables such as citric acid (R Spearman: 0.309; *p*: 0.017), *KLK3* (R Spearman: −0.412; *p*: 0.001), and *PCA3* (R Spearman: 0.556; *p* < 0.0001), similarly to what occurred with PSA_blood (R Spearman: 0.760; *p* < 0.0001), although it indicated a weaker correlation between age and sEV transcript levels in semen than the one estimated for age and PSA in blood. No correlation was found when considering only prostate-affected individuals.

### 2.5. Evaluation of the Potential of Semen Variables as PCa Diagnostic Biomarkers

Subsequently, we compared the expression profile of proteins or sEV-mRNAs in semen between malignant and non-malignant samples with potential implications for diagnosis.

We found that only the expression values of sEV-contained *PCA3* gene resulted in good and statistically significant predictive accuracy (AUC: 0.796; *p*-value < 0.0001; Sn: 44.4%; Sp: 90.6%) for discriminating between non-malignant (HCt+BPH) and malignant samples. As a comparison, a ROC curve analysis of blood PSA was also performed (AUC: 0.884; *p*-value < 0.0001; Sn: 77.8%; Sp: 84.4%) (Table 2A).

With the purpose of distinguishing PCa from benign prostate conditions such as BPH, to increase clinical relevance, the same analysis was performed in samples from individuals with blood PSA > 4 ng/mL. Similarly to what was obtained when using blood PSA, no mRNA in semen sEVs was able to distinguish between phenotypes (Table 2B). Thus, a multivariate logistic regression analysis was performed in order to determine whether a multiplex molecular model could reflect the malignancy of prostate disease. We obtained a model combining the expression values of *CCNQ* and *DUSP23* in semen sEVs that can distinguish between BPH and PCa samples (AUC: 0.722; *p*-value: 0.059; Sn: 100%; Sp: 25%). The accuracy increased when blood_PSA and/or citric acid was included in the analysis, resulting in the models comprising PSA+*PCA3+CREB3L4+CCNQ+DUSP23* (AUC: 0.806; *p*-value: 0.010; Sn: 96.3%; Sp. 25%) and PSA+citric acid+*PCA3+CREB3L4+CCNQ+DUSP23* (AUC: 0.866; *p*-value: 0.002; Sn: 96.3%; Sp. 50%), much more useful than PSA or individual transcripts for diagnosis (Table 2B). Interestingly, the combination of blood PSA and semen sEV-*DUSP23* levels was found to be able to discriminate clinically significant (PCa GS 7+8) from non-significant (BPH + PCa GS6) samples (AUC: 0.721; *p*-value: 0.029; Sn: 71.4%; Sp: 85.7%) (Table 2D).

### 2.6. Evaluation of the Potential of Semen Variables as PCa Prognostic Biomarkers

The same type of analysis was performed in order to determine whether a multiplex model could reflect the severity or degree of PCa affectation, which is associated with the expected outcome of the disease. We performed the analysis considering only individuals with prostate disease (excluding the HCt group). When samples were staged into prognostic groups (I, IIA, IIB, IIC, and IIIB in accordance with the AJCC PCa staging system), the combined RNA model (*PCA3+CREB3L4+DUSP23*) was able to discriminate between BPH/low/intermediate-risk tumors (BPH+PCaI+IIA+IIB) and high-risk tumors (PCa IIC+IIIB) (AUC: 1), similarly to the model including PSA_blood (PSA_blood+*PCA3+CREB3L4+DUSP23+KLK3*) (Table 2E).

All these results indicate that a seminal EV gene panel alone or in combination with PSA in blood might be useful to identify aggressive PCa in patients with moderately elevated levels of PSA.

## 3. Discussion

Based on studies from recent decades, gene expression profiles have become the basis for improving the diagnosis and prognosis of cancer including PCa, with promising clinical use. With the emergence of high-throughput technologies, large amounts of data have been generated, mainly in tissue, with the aim of enhancing prediction of the presence of tissue malignancy, its behavior and a precise stage classification. Some publications have integrated all this metadata on tissue to identify RNA-based biomarkers that help to determine high-grade prostate cancer patients with a higher risk of progression (https://doi.org/10.1101/2024.08.21.608965). While tissue biopsies remain the gold standard for tumor diagnosis, liquid biopsies offer a less invasive but valuable complementary alternative for diagnosis and monitoring the progression of the disease. Interest in using semen as a liquid biopsy of the prostate is increasing and reinforced by the origin of seminal fluid secretions providing supplementary evidence about dysregulated pathways of a heterogeneous tumor. Thus, the goal of the current study is to explore the potential of promising selected transcripts as classifiers for PCa in a non-invasive manner such as those in semen EVs. Specifically, single and multivariate models, including other prostate-specific molecules in semen such as PSA and citric acid, are evaluated to determine their prediction capacity to discriminate PCa from BPH samples as well as between different PCa risk levels in those patients with moderately increased PSA levels (4–10 ng/mL) where accurate biomarkers are more necessary.

Referring to PCa-related molecules in semen, our results provide evidence that PSA in semen, contrary to blood_PSA, is not a reliable indicator of individuals with PCa compared to healthy and/or BPH individuals. Additionally, concerning citric acid, we observed statistically decreased levels of this molecule in semen in PCa compared to healthy controls, but no difference was found related to the severity of the disease.

Interestingly, we show alterations of gene expression for sEV-contained *CREB3L4*, *CREB3L2*, *CCNQ*, *DUSP23*, *PCA3,* and *KLK3* in different PCa clinical conditions. Specifically, *PCA3* and *KLK3* are dysregulated in PCa compared to HCt controls, whereas *CREB3L4*, *CREB3L2*, *CCNQ,* and *DUSP23* are dysregulated in high-risk compared with low-risk PCa. These genes are altered to different degrees between the two conditions; thus, an additional model led us to obtain better results for PCa biomarkers, not only for diagnosis but most importantly for predicting the evolution of the PCa disease.

Our results are in line with the function of the selected coding and non-coding RNAs. *PCA3* is classified as a long non-coding RNA (lncRNA), RNA transcripts that participate in the regulation of gene expression, influencing cell growth, programmed cell death, metastasis, and resistance to treatments through pathways like PI3K/AKT, WNT/β-catenin, and androgen receptor signaling. Specifically, *PCA3* is overexpressed in PCa cells, playing a significant role in PCa progression, with potential as a diagnostic marker. Particularly, *PCA3* has been shown to promote the proliferation of prostate cancer cells through modulating androgen receptor (AR) signaling [13].

PSA, coded by *KLK3*, is also involved in angiogenesis, potentially contributing to tumor growth and metastasis. The androgen receptor (AR) plays a crucial role in regulating *KLK3* expression [14], and AR signaling pathways are often dysregulated in prostate cancer.

CREB3L4 is linked to cell proliferation [15] through HDAC3; to DNA damage repair by interplaying with AKT1, AKT3, and CHD1; and to castration resistance through interactions with CREBBP, EP300, and AKT2 (https://doi.org/10.1101/2024.08.21.608965).

CCNQ is related to DNA damage repair through CDK12. DUSP23 is related to advanced PCa, which is prone to metastatic progression. It is suggested to have a role in the metastatic spread of cancer cells to bone tissue through its interaction with ITGB4 (https://doi.org/10.1101/2024.08.21.608965). *DUSP23* upregulation has been associated with poor prognosis in patients with other types of cancer, such as lung cancer and breast cancer [16].

Distinguishing tumors from normal samples is crucial in cancer research and diagnosis, but most importantly, the early and accurate detection of prostate cancer versus benign prostatic hyperplasia (BPH) is critical for reducing unnecessary biopsies. Similarly to what occurs for blood_PSA, clear differences in gene expression levels are obtained when control and PCa phenotypes are compared, but not when comparing malignant and non-malignant prostate-affected individuals, suggesting that BPH-related dysregulation shares changes in transcript levels with PCa, with enough overlap to prevent accurate separation. This phenomenon results in at least half of BPH cases being misclassified by the individual gene model. Our results showed that long RNAs have the potential to be used as PCa biomarkers in sEV semen samples when applied in a multivariable test, also including blood_PSA and citric acid, capable of distinguishing individuals affected by PCa from those affected by BPH, thus contributing to the precise prediction of PCa, especially in those cases where PSA tests fail to accurately predict disease. Specifically, our combined RNA models (*CCNQ+DUSP23*: AUC: 0.722), PSA_blood+RNA model (AUC: 0.806), and PSA_blood+ citric acid+ RNA models (AUC: 0.866) in semen resulted in a better AUC than the one obtained from the commercially available Progensa PCA3 assay in urine samples (described AUC: 0.685), suggesting their potential for clinical application.

In the same way, predicting the outcome of the disease may lead to decreasing overtreatment. Combined models have been described to be useful as biomarkers for disease. The sensitivity and specificity of screening methods are crucial for the diagnosis and prognosis of cancer. Some methods are recommended due to their superior sensitivity and negative predictive value compared to conventional screening. In our case, semen sEV-RNA biomarker combinations can improve the diagnostic value for aggressive prostate cancer in patients within the 4–10 ng/mL PSA range. In line with this, the models that we obtained after multivariate regression analysis present an AUC > 0.7, which is generally considered clinically useful. Fortunately, PCa is a slow-growing tumor; thus, the diagnosis of PCa not only involves the detection of cancerous cells but also the determination of its true aggressiveness. The use of sEV RNA-based tests in semen can help reduce the proportion of unnecessary biopsies by better discrimination of aggressive from nonaggressive cancers while missing a reduced number of clinically significant cancers.

Some tissue-based molecular tests are available in the market with the aim of estimating the likely course of the disease. Decipher^®^ and Oncotype Prostate^®^ tests are used to complement traditional clinical diagnostic features. Decipher results are based on the expression behavior of 22 genes, whereas Oncotype relies on the expression behavior of 17 genes in biopsy/prostatectomy prostatic tissue.

Our proposal is based on non-invasive prostate cancer-specific biomarkers to complement PSA-related cancer screening and distinguish between the aggressive prostate cancer tumor type and the indolent prostate cancer form before biopsy. In this way, overdiagnosis of PCa could be reduced and individuals requiring prostate biopsy confirmation could be more accurately selected.

Current non-invasive screening methods for PCa, mainly in urine, are limited in their ability to detect the disease at an early stage, assess tumor aggressiveness, or predict its progression. Consequently, identifying PCa-specific EV-contained RNAs released into semen during disease development may offer a valuable approach for early diagnosis. These sEV-RNAs could also help predict disease progression and guide treatment strategies, ultimately improving patient outcomes. Overall, our findings present strong evidence that differentially expressed RNAs in semen EVs serve as effective biomarkers for improving the diagnosis of PCa and assessing its severity with high diagnostic accuracy. In conclusion, our results support the potential of semen sEV-based RNA profiling in the clinical management of prostate cancer.

Previous studies from our group have identified both quantitative and qualitative alterations in the small RNA, such as the microRNA (miRNA), composition of semen sEVs from PCa patients [11]. By studying the miRNA expression levels within these vesicles, we obtained a predictive model incorporating the expression of several miRNAs—particularly miR-142-3p, miR-142-5p, and miR-223-3p—which proved effective in detecting the presence of prostate cancer cells and distinguishing PCa from benign conditions such as benign prostatic hyperplasia. This suggests that such a molecular test could serve as a valuable non-invasive biomarker complement to PSA analysis in diagnosing prostate cancer [11].

Comparison of the performance of classifiers based on different profiles is necessary. More interestingly, the integration of different types of RNA biomarkers including (current) long and (previous) small RNA biomarkers may enable a more precise non-invasive diagnosis and/or prognosis of PCa disease.

To summarize, in the context of PCa screening, there is a pressing need for new, more effective non-invasive biomarkers with strong prognostic abilities to evaluate the risk of prostate cancer progression at the time of diagnosis, in an early stage of the disease. We describe that classifiers based on combined long transcript levels in semen sEVs have clinical accuracy to distinguish patients with higher GS and AJCC stage, suggesting their potential as indicators of disease progression rather than early detection. These multivariate modes hold significant potential for clinical use, allowing a more comprehensive assessment, increasing diagnostic accuracy, and helping with accurate clinical decision-making.

## 4. Materials and Methods

### 4.1. Subjects of Study

Patients and controls participating in the study were selected from men referred to the Urology Service of the Bellvitge Hospital and the Andrology Service of Fundació Puigvert. The ethical review board of both centers approved the study, and a written informed consent document was signed by all the participants. Semen specimens were collected from 35 individuals, selected among those who consulted for a PCa diagnosis, showed moderately elevated blood PSA levels (4–18 ng/mL), and consented to undergo a prostate biopsy. Specifically, this group comprised 27 men with biopsy-confirmed PCa, including those who had previously undergone vasectomy [PCa-V, n = 7; age (years): 58 ± 9.70] and non-vasectomized individuals [PCa-noV, n = 20; age (years): 59.45 ± 6.48], and additionally, 8 non-vasectomized individuals with benign prostatic hyperplasia (BPH) or prostate enlargement [control group 2; age (years): 58.63 ± 4.50] who presented with elevated PSA levels (>4 ng/mL) but no detectable cancer on biopsy. Additionally, 17 vasectomized men [HCt_V; age (years): 40 ± 3.95] and 7 normozoospermic non-vasectomized individuals [HCt_noV; age (years): 40.57± 2.94] who consulted for vasectomy or infertility (inclusion criteria: individuals over 30 years of age) were defined as the healthy control group (Table 1).

After the histopathological evaluation of prostate biopsies, samples were classified into low-grade (GS6), intermediate-grade (GS7), and high-grade (≥GS8) cancer. Samples were also stratified by tumor class according to the 8th edition of the AJCC prognostic groups, which consider TNM staging, pre-treatment PSA levels, and tumor Gleason grade, the latter being of particular importance. The classifications comprised I (low risk, including organ-confined GS6 samples with PSA < 10 ng/mL), IIA (low–intermediate risk, consisting of organ-confined GS6 samples with PSA > 10 ng/mL), IIB (intermediate risk, which includes organ-confined GS7 (3 + 4) samples), IIC (intermediate–high risk, comprising organ-confined GS7 (4 + 3) samples), and IIIB (high-risk–advanced cases characterized by extra-prostatic tumor extension, involving cT3 samples of any GS grade group and PSA levels) (Table 1).

### 4.2. PSA Determination by Electrochemiluminescent Immunoassay in Semen Samples

PSA concentrations in semen were successfully quantified in 51 of the 59 samples included in the study. The remaining 8 samples were not analyzed due to sample volume limitations. Considering that PSA concentrations in semen can be up to 100,000 times higher than in serum, a 1/1000 dilution of the samples was prepared in two sequential steps (1/20 followed by 1/50) using PBS as a diluent. Subsequently, the samples were further diluted (1/50) in Universal Diluent (Roche; Basel, Switzerland) as a preparatory step for quantification with the Elecsys Total PSA assay (Roche; Basel, Switzerland) on the Roche Diagnostics Cobas 8000 e801 analyzer (Roche; Basel, Switzerland).

### 4.3. Citric Acid Quantification by Photometry in Semen Samples

The amount of citric acid in seminal plasma was measured with CitricScreen (ScopeScreen; Brighton, MI, USA) as detailed by the manufacturer. Citric acid in semen is linked to prostate health: for instance, low levels of citric acid are described to be associated with subclinical prostatitis and prostate cancer [7].

### 4.4. Semen Sample Processing and sEV Isolation

Semen samples were obtained by masturbation after a period of 3–5 days of sexual abstinence. Samples were liquefied for 30 min at 37 °C, and were submitted to two rounds of centrifugation (1600× *g* for 10 min, then 16,000× *g* for 10 min at 4 °C) in order to remove cells, cellular bodies and apoptotic bodies. Collected seminal plasma was immediately stored at −80 °C until needed. sEVs were isolated by filtration through a 0.22 µm filter and ultracentrifugation (100,000× *g* for 2 h at 4 °C) as previously described [11,17]. The pellet was resuspended in 100 µL PBS and samples were stored at −80 °C. Nanoparticle tracking analysis was conducted using NanoSight NS300 (Malvern Instruments Ltd.; Malvern, UK).

### 4.5. Total RNA Isolation

Total RNA was extracted from sEV suspension samples using the miRNeasy Micro Kit (Qiagen; Venlo, The Netherlands) as previously described [11,18]. RNA concentration was measured using the Quant-iT RNA Assay Kit (5–100 ng/µL) (Invitrogen, Carlsbad, CA, USA) with the QUBIT fluorometer. To assess RNA quality, the OD 260/280 nm ratio was determined for all samples using a Nanodrop UV-Vis spectrophotometer (Thermo Fisher Scientific, Waltham, MA, USA), ensuring values ≥ 1.65.

### 4.6. Transcript Detection by RT-qPCR

Reverse transcription (RT) was performed on 50 ng of semen sEV total RNA in a final volume of 20 µL using the SuperScript^®^ IV First-Strand cDNA Synthesis Reaction kit (Invitrogen-Thermo Fisher Scientific; Waltham, MA, USA) with 50 µM of random hexamers.

Primer pairs for the detection of selected transcripts were designed using the PRIMER3 tool (https://primer3.ut.ee) (accessed in 17 September 2024). The designed primers were then evaluated for specificity and efficiency before being used. Sequences of primers are described in Table 3.

cDNA samples were diluted (2X) for quantitative real-time PCR (qPCR) analysis, and 10 µL PCR reactions were performed (in duplicates) with 1 µL (*RPL19*, *RPS17*, *CCNQ*, *DUSP23*, *MXD4*, *KLK3*) or 2 µL (*CREB3L4*, *CREB3L2*) of diluted cDNA, 5 µL of 2X SYBR Green mix (Roche; Basilea, Switzerland), and 500 nM of forward and reverse primers. Cycling conditions were set up in an LC96 (LightCycler^®^ 96 Real-time PCR Instrument; Roche; Basilea, Switzerland) as follows: 5 min at 95 °C, followed by 40 cycles consisting of 10 s at 95 °C, 10 s at 56 °C, and 15 s at 72 °C, and finally melting curve analysis at 65 °C to 97 °C. To correct for overall differences between samples, target gene expression was calculated relative to the expression of the genes *RPL19* and *RPS17* [21]. The expression behavior of these genes in our semen EV sample cohort was additionally checked, showing that the means of *RPL19* and *RPS17* expression values present a low coefficient of variation among samples (Appendix A). The relative quantification (RQ) values were calculated using the 2dCq strategy.

### 4.7. Statistical Analysis

Given that our sample size is moderately limited, the non-parametric Mann–Whitney U-test was used to evaluate differences in relative expression of selected transcripts between groups. A Spearman correlation analysis was performed to assess a potential relationship between sEV transcript levels and age (as a confounding factor). Assessment of the ability to differentiate samples indicating prostate tumor malignancy or the aggressiveness of the disease was performed by receiver operating characteristic (ROC) curve analysis of the RQ values. Accuracy was measured as the area under the ROC curve (AUC). The threshold value was determined by Youden’s index, calculated as sensitivity plus specificity-1. A multivariate binary logistic regression (backward stepwise conditional method) was used for selection of the optimal combination of variables associated with the presence or severity of the disease. The binary logistic regression model provides the following estimation of the Logit function:Logit(p) = B0 + B1X1 + B2X2 + …
where p = P (presence of prostate cancer), Logit (p) = log(p/(1 − p)) = log(odds), B = logOR and Xn = the expression values of the RNAs. Therefore, if we use this estimated model as a prediction model, with the standard classification cutoff of 0.5, we can classify individuals with a positive Logit function estimation as “positive for PCa” and individuals with a negative Logit function estimation as “negative for PCa”.

All data analyses were performed using SPSS software (version 15.0; SPSS Inc.; IBM; Chicago, IL, USA). A *p*-value ≤ 0.05 was considered significant.

## Figures and Tables

**Figure 1 ijms-26-09562-f001:**
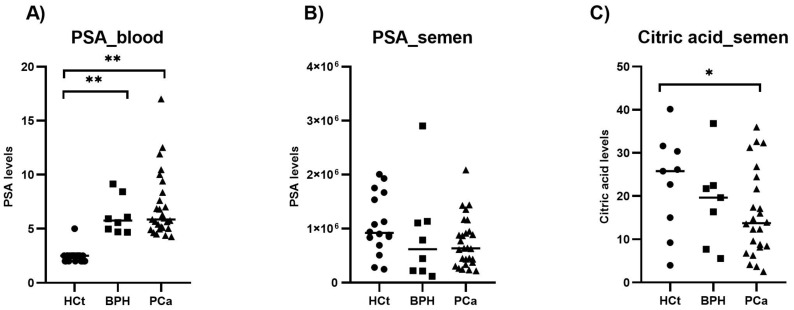
PSA and citric acid levels in semen samples compared to PSA levels in blood. The levels of PSA (ng/mL) (**B**) and citric acid (mg/mL) (**C**) were quantified in semen samples from healthy controls (HCt), benign prostate hyperplasia (BPH), and prostate cancer (PCa) individuals and compared with PSA levels in blood (**A**). The horizontal bar displays the median expression level. Significant differences between groups are indicated: * *p*-value < 0.05; ** *p*-value < 0.01 (Mann–Whitney U-test).

**Figure 2 ijms-26-09562-f002:**
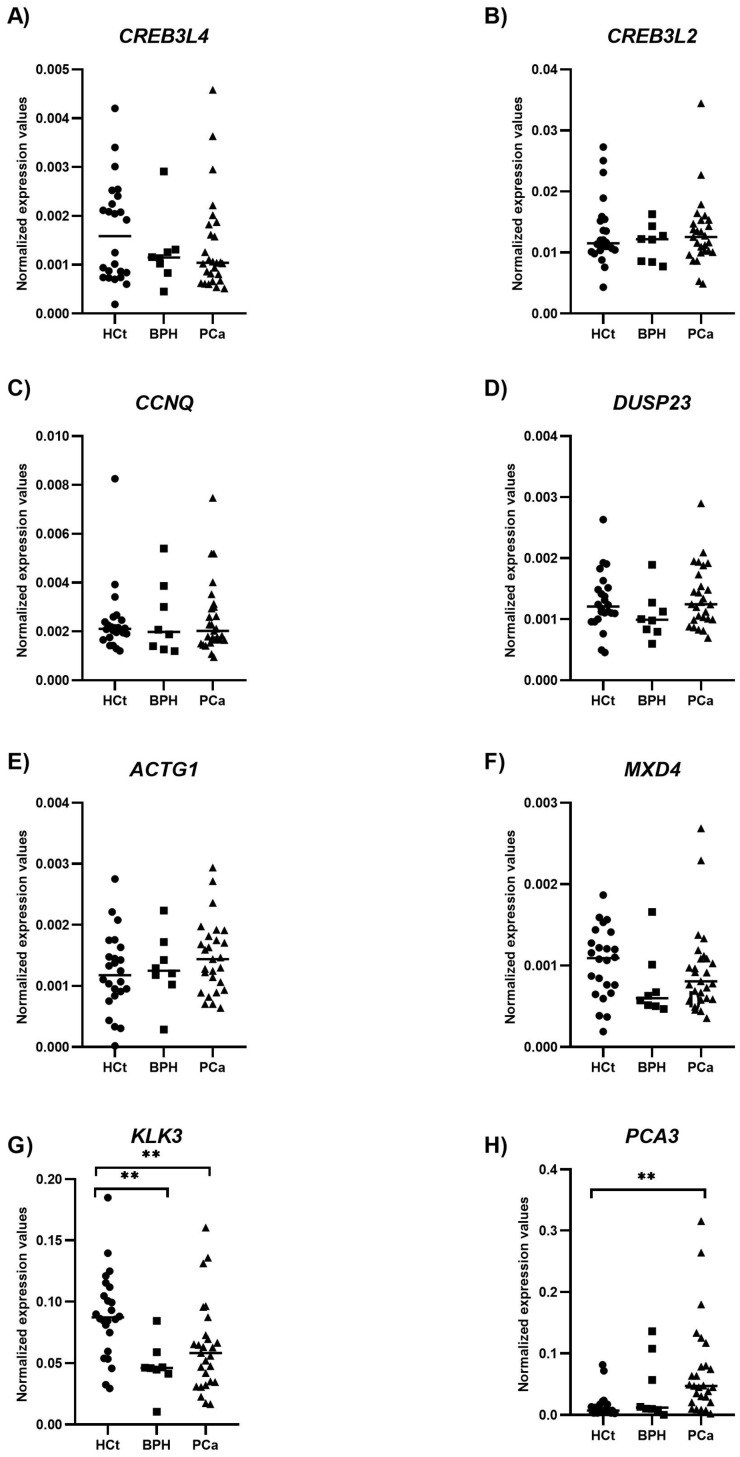
Semen sEV RNA levels in prostate cancer samples compared to controls. Expression profiling of the sEV transcript levels [*CREB3L4* (**A**), *CREB3L2* (**B**), *CCNQ* (**C**), *DUSP23* (**D**) *ACTG1* (**E**), *MXD4* (**F**), *KLK3* (**G**), and *PCA3* (**H**)] from semen of healthy controls (HCt), benign prostate hyperplasia (BPH), and prostate cancer (PCa) individuals. The horizontal bar displays the median expression level. Significant differences between groups are indicated: ** *p*-value < 0.01 (Mann–Whitney U-test).

**Figure 3 ijms-26-09562-f003:**
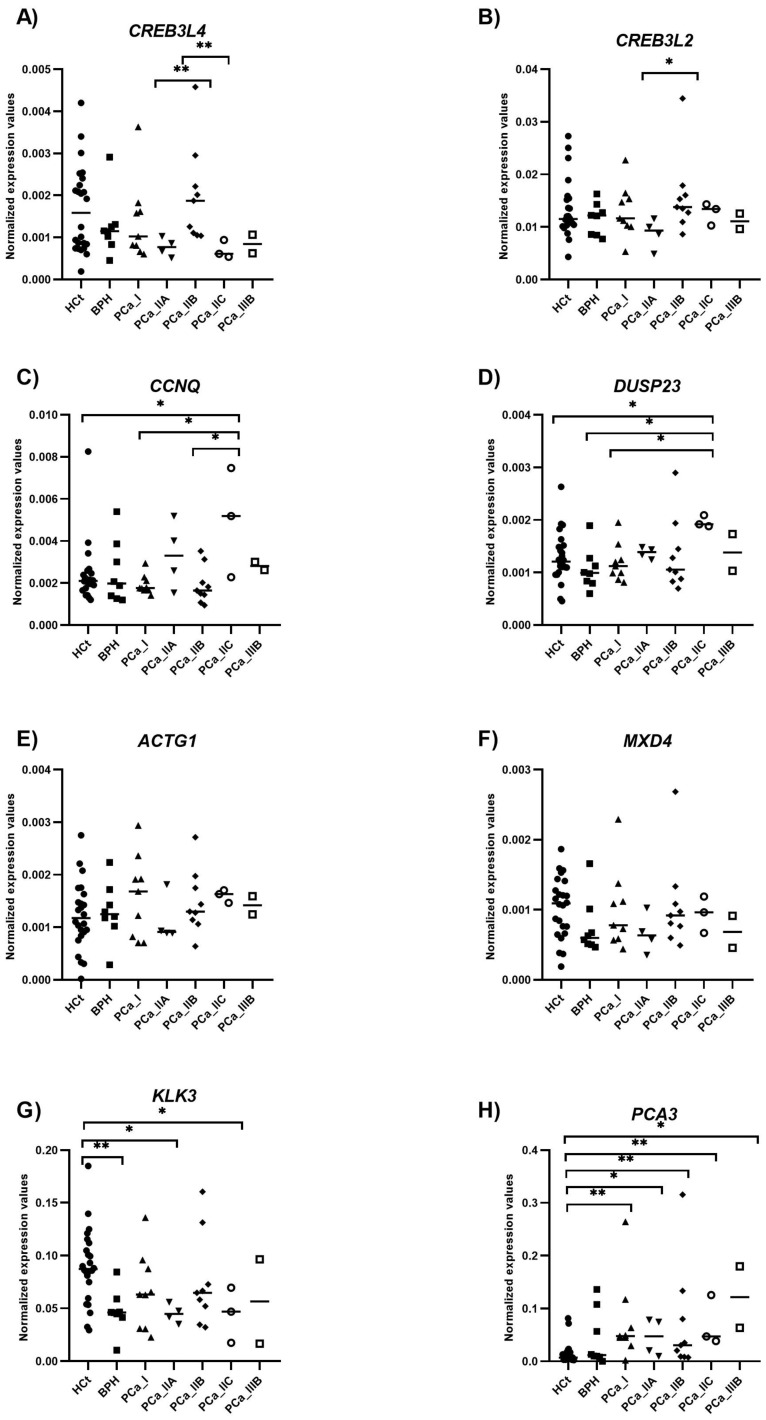
Semen sEV RNA levels in clinically staged PCa samples by prognostic groups. Expression profiling of the sEV transcript levels [*CREB3L4* (**A**), *CREB3L2* (**B**), *CCNQ* (**C**), *DUSP23* (**D**) *ACTG1* (**E**), *MXD4* (**F**), *KLK3* (**G**), and *PCA3* (**H**)] from semen is shown. The horizontal bar displays the median expression value. Significant differences between groups are indicated: * *p*-value < 0.05; ** *p*-value < 0.01 (Mann–Whitney U-test). HCt: healthy controls; BPH: benign prostatic hyperplasia. PCa samples were staged into prognostic groups in accordance with the 8th Edition of AJCC (American Joint Committee on Cancer) staging system for PCa: from the lowest-risk (PCa I) to highest-risk (PCa IIIB) tumors.

**Table 1 ijms-26-09562-t001:** Clinical data of individuals included in the study.

Variable	HCt_noV	HCt_V	BPH	PCa_noV	PCa_V
Total (*n*)	7	17	8	20	7
Age, range (years)	37–44	32–48	52–64	50–72	43–68
Age, mean ±SD (years)	40.57 ± 2.94	40 ± 3.95	58.63 ± 4.50	59.45 ± 6.48	58 ± 9.70
**Pre-biopsy PSA**					
≤10 ng/mL (*n*)	nd	nd	8	19	4
>10 ng/mL (*n*)	nd	nd	0	1	3
Pre-biopsy PSA, mean ±SD (ng/mL)	nd	nd	6.19	6.43	8.97
**Gleason score**					
Gleason score biopsy 6(3+3) (*n*)	nd	nd	nd	10	3
Gleason score biopsy 7(3+4) (*n*)	nd	nd	nd	7	4
Gleason score biopsy 7(4+3) (*n*)	nd	nd	nd	2	0
Gleason score biopsy 8(4+4) (*n*)	nd	nd	nd	1	0
**Clinical stage**					
T1c (*n*)	nd	nd	nd	14	2
T2a (*n*)	nd	nd	nd	5	1
T2c (*n*)	nd	nd	nd	0	3
T3a (*n*)	nd	nd	nd	1	1
**Prognostic group ***					
I (*n*)	nd	nd	nd	7	2
IIA (*n*)	nd	nd	nd	3	1
IIB (*n*)	nd	nd	nd	6	3
IIC (*n*)	nd	nd	nd	3	0
IIIB (*n*)	nd	nd	nd	1	1

HCt_noV: healthy control non-vasectomized individuals; HCt_V: healthy control vasectomized individuals; BPH: benign prostate hyperplasia group; PCa_noV: prostate cancer from non-vasectomized individuals; PCa_V: prostate cancer from vasectomized individuals. * American Joint Committee on Cancer Prognostic Stage grouping (8th edition). nd: non-determined.

**Table 2 ijms-26-09562-t002:** Receiver operating characteristic (ROC) analysis showing the predictive efficiency of RNAs in seminal small extracellular vesicles (sEVs) for PCa diagnosis.

Markers	AUC (*p*-Value)	95% CI	Sn %	Sp %	PPV %	NPV %
**A.** (HCt + BPH) vs. PCa						
*CREB3L4*	0.576 (0.319)	0.428–0.724	11.1	90.6	50	54.71
*CREB3L2*	0.522 (0.773)	0.372–0.672	3.7	100	100	55.17
*CCNQ*	0.504 (0.958)	0.351–0.657	3.7	96.9	50	54.38
*DUSP23*	0.578 (0.308)	0.430–0.725	25.9	81.3	53.84	56.52
*ACTG1*	0.593 (0.221)	0.447–0.740	40.7	75	57.89	60
*MXD4*	0.549 (0.523)	0.400–0.697	0	100	0	54.23
*KLK3*	0.634 (0.078)	0.489–0.779	48.1	65.6	54.16	60
*PCA3*	0.796 (**0.000 ***	0.677–0.915	59.3	84.4	76.19	71.05
*PCA3/KLK3*	0.808 (**0.000 ***)	0.688–0.927	44.4	90.6	80	65.9
PSA_blood	0.884 (**0.000 ***)	0.797–0.970	77.8	84.4	84	81.81
PSA_semen	0.550 (0.513)	0.399–0.701	0	100	0	54.23
Citric acid_semen	0.633 (0.080)	0.487–0.779	25.9	68.8	41.17	52.38
**B.** BPH vs. PCa						
*CREB3L4*	0.472 (0.814)	0.252–0.692	100	0	77.14	0
*CREB3L2*	0.593 (0.432)	0.365–0.820	100	0	77.14	0
*CCNQ*	0.525 (0.829)	0.268–0.783	100	0	77.14	0
*DUSP23*	0.708 (0.077)	0.502–0.915	100	0	77.14	0
*ACTG1*	0.569 (0.556)	0.352–0.787	100	0	77.14	0
*MXD4*	0.648 (0.209)	0.425–0.871	100	0	77.14	0
*KLK3*	0.634 (0.255)	0.435–0.833	100	0	77.14	0
*PCA3*	0.644 (0.223)	0.401–0.886	100	0	77.14	0
*PCA3/KLK3*	0.634 (0.255)	0.392–0.877	100	0	77.14	0
PSA_blood	0.563 (0.596)	0.349–0.776	100	0	77.14	0
PSA_semen	0.560 (0.610)	0.288–0.832	100	0	77.14	0
Citric acid_semen	0.433 (0.569)	0.192–0.674	100	0	77.14	0
Combined RNA model (*CCNQ*+*DUSP23*)	0.722 (0.059)	0.521–0.923	100	25	81.81	100
Combined PSA_blood+RNA model (PSA_blood+*PCA3*+*CREB3L4*+*CCNQ*+*DUSP23)*	0.806 (**0.010**)	0.647–0.964	96.3	25	66.66	81.25
Combined PSA+citric+RNA model (PSA_blood+citric acid+*PCA3*+*CREB3L4*+*CCNQ*+*DUSP23*)	0.866 (**0.002**)	0.742–0.989	96.3	50	86.66	80
**C.** (HCt + BPH+PCa_GS6) vs. (PCa GS7+GS8)						
*CREB3L4*	0.532 (0.722)	0.359–0.704	0	100	0	76.27
*CREB3L2*	0.616 (0.193)	0.455–0.777	7.1	100	100	77.58
*CCNQ*	0.520 (0.824)	0.324–0.716	0	100	0	76.27
*DUSP23*	0.600 (0.262)	0.409–0.791	7.1	97.8	50	77.19
*ACTG1*	0.613 (0.203)	0.463–0.764	0	100	0	76.27
*MXD4*	0.508 (0.929)	0.348–0.668	0	100	0	76.27
*KLK3*	0.568 (0.444)	0.391–0.746	0	100	0	76.27
*PCA3*	0.717 (**0.015**)	0.572–0.863	14.3	97.8	66.66	78.57
*PCA3/KLK3*	0.716 (**0.015**)	0.552–0.880	7.1	97.8	50	77.19
PSA_blood	0.832 (**0.000 ***)	0.724–0.939	28.6	88.9	44.44	80
PSA_semen	0.537 (0.682)	0.373–0.700	0	100	0	76.27
Citric acid_semen	0.538 (0.669)	0.384–0.693	0	100	0	76.27
Combined RNA model (*PCA3+CREB3L4+KLK3+CCNQ+DUSP23*)	0.746 (**0.006**)	0.573–0.919	35.7	97.8	83.33	83.01
Combined PSA_blood+RNA model (PSA_blood+*PCA3+CREB3L4+KLK3+DUSP23*)	0.906 (**0.000 ***)	0.831–0.982	64.3	91.1	69.23	89.1
Combined PSA+citric+RNA model (PSA_blood+citric acid+*PCA3*+*DUSP23*)	0.881 (**0.000 ***)	0.777–0.984	64.3	93.3	75	89.36
**D.** (BPH + PCa_GS6) vs. (PCa_GS7+GS8)						
*CREB3L4*	0.604 (0.304)	0.406–0.801	14.3	90.5	50	61.29
*CREB3L2*	0.658 (0.117)	0.476–0.841	21.4	85.7	50	62.06
*CCNQ*	0.522 (0.827)	0.316–0.728	7.1	100	100	61.76
*DUSP23*	0.639 (0.167)	0.441–0.838	42.9	90.5	75	70.37
*ACTG1*	0.578 (0.439)	0.385–0.772	7.1	95.2	50	60.6
*MXD4*	0.612 (0.266)	0.420–0.804	7.1	90.5	33.33	59.37
*KLK3*	0.592 (0.363)	0.388–0.796	14.3	95.2	66.66	62.5
*PCA3*	0.558 (0.567)	0.357–0.758	14.3	95.2	66.66	62.5
*PCA3/KLK3*	0.575 (0.459)	0.372–0.778	7.1	95.2	50	60.6
PSA_blood	0.646 (0.148)	0.451–0.842	28.6	85.7	57.14	64.28
PSA_semen	0.514 (0.893)	0.311–0.717	0	100	0	60
Citric acid_semen	0.612 (0.266)	0.422–0.803	21.4	90.5	60	63.33
Combined RNA model (*PCA3+DUSP23*)	0.670 (0.092)	0.469–0.871	50	85.7	70	72
Combined PSA_blood+RNA model (PSA_blood+*DUSP23)*	0.721 (**0.029**)	0.534–0.908	71.4	85.7	76.92	81.81
**E.** [BPH + PCa_I + PCa_IIA+PCa_IIB] vs. [PCa_IIC + PCa_IIIB]						
*CREB3L4*	0.844 (0.052)	0.680–1.000	0	100	0	91.42
*CREB3L2*	0.552 (0.768)	0.317–0.787	0	100	0	91.42
*CCNQ*	0.865 (**0.039**)	0.667–1.000	33.3	100	100	94.11
*DUSP23*	0.917 (**0.018**)	0.820–1.000	0	96.9	0	91.17
*ACTG1*	0.656 (0.377)	0.493–0.820	0	100	0	91.42
*MXD4*	0.635 (0.444)	0.391–0.880	0	100	0	91.42
*KLK3*	0.583 (0.637)	0.244–0.922	0	100	0	91.42
*PCA3*	0.615 (0.517)	0.380–0.849	0	100	0	91.42
*PCA3/KLK3*	0.719 (0.216)	0.497–0.940	0	100	0	91.42
PSA_blood	0.604 (0.556)	0.423–0.786	0	100	0	91.42
PSA_semen	0.688 (0.289)	0.301–1.000	0	100	0	91.42
Citric acid_semen	0.594 (0.596)	0.198–0.989	0	100	0	91.42
Combined RNA model (*PCA3*+*CREB3L4+DUSP23*)	1.000 (**0.005**)	1.000–1.000	100	100	100	100
Combined PSA_blood+RNA model (PSA_blood+*PCA3*+*CREB3L4+DUSP23+KLK3*)	1.000 (**0.005**)	1.000–1.000	100	100	100	100

AUC: area under the curve; Sn: sensitivity; Sp: specificity; PPV: positive predictive value; NPV: negative predictive value. *p*-value < 0.05 is depicted in bold; * *p*-value < 0.0001

**Table 3 ijms-26-09562-t003:** Sequences of primers used for the detection of selected transcripts.

Gene	NCBI Transcript Name	Primer Forward	Primer Reverse	Amplicon (nt)
*CREB3L4*	NM_001255978.2	AAACCCTGTTCCTGACCGAT	ACCAAGGAGATGTTGTGCCT	268
*CREB3L2*	NM_194071.4	CACTGGGGTTGATTCCTCGTG *****	AATGCAGGTGGTCCACTGGG *****	116
*CCNQ*	NM_152274.5	ATCATGGAGGCAGGTGTCAA	GTAAGGGTCATAGGCGTCCA	111
*DUSP23*	NM_001319658.2	GGAGAGGCTGTGGGAGTG	CGTAGTCGTCGGATTTCAGC	125
*ACTG1*	NM_001199954.3	GGAACAAAAGGCGGGGTC	ATGGGGTACTTCAGGGTCAG	233
*MXD4*	NM_006454.3	GAAAAGCACAGACGAGCCAA	GTCCTGCTCCTCCAGTTTCT	144
*KLK3*	NM_001648.2	TTGTCTTCCTCACCCTGTCC	ACGCTTTTGTTCCTGATGCA	202
*PCA3*	NR_015342.2	GCACATTTCCAGCCCCTTTA **^#^**	GGCATTTCTCCCAGGGATCT **^#^**	78
**Normalizer**	**NCBI Transcript Name**	**Primer Forward**	**Primer Reverse**	**Amplicon (nt)**
*RPL19*	NM_000981.4	GCACATGGGCATAGGTAAGC	CAGGCTGTGATACATGTGGC	148
*RPS17*	NM_001021.6	CGCCATTATCCCCAGCAAAA	CTGCCGAAGTCCAAAAGCTT	225

All primers were designed with the PRIMER3 tool, with the exception of the primer pairs marked with * [19] and ^#^ [20].

## Data Availability

Data is contained within the article and Appendix A.

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
