# Peer review of "Candidate Transcript Panel in Semen Extracellular Vesicles Can Improve Prediction of Aggressiveness of Prostate Cancer"

_ijms, 2025, doi:10.3390/ijms26199562_

Round 1

Reviewer 1 Report

Comments and Suggestions for Authors

1、The statistical analysis section (Lines 453–471) does not specify the software settings or assumptions (e.g., normality tests, multiple comparison corrections) used for the Mann-Whitney U-test and ROC curve analyses, which may affect reproducibility.

2、 Does the expression of long transcripts in semen sEVs vary due to other clinical factors in prostate cancer patients (e.g., age, treatment history, or prostate inflammation status)? It is recommended to include covariance analysis to rule out secondary confounding factors.

3、 The manuscript interchangeably uses terms such as “semen sEVs,” “seminal sEVs,” and “sEV-contained transcripts” without clear definition or justification, which may cause confusion for readers.

4、 Lines 29–31: “Our findings also present strong evidence that classifiers based on combined long transcript levels in semen sEVs serve as effective biomarkers, alone or in combination with blood PSA and/or semen citric acid levels, for improving diagnosing PCa and assessing its severity and disease progression with high accuracy.” This sentence contains multiple clauses and concepts (classifiers, biomarkers, combinations, diagnosis, and prognosis), making it verbose

5、 Line 410: “…two rounds of centrifugation (1600x g for 10 min, then 16,000x g for 10 min at 4 ºC) were undergone in order to remove cells…” (past tense, but “were undergone” is inappropriate).

6、 Line 43: “…revealing patients with a suspicion of cancer.” The phrase “a suspicion” is inappropriate, as “suspicion” is typically an uncountable noun in this context.

7、 Line 393: “…which prevented from a proper quantification.” The phrase “prevented from a proper quantification” lacks an article and is poorly worded.

8、 Line 112: “After revising the list of their candidate biomarkers, we selected CREB3L4. and its family member CREB3L2…” The phrase “the list of their candidate biomarkers” can be simplified.

9、 Line 53: “4-10 ng/ml” (inconsistent with “ng/mL” used elsewhere, e.g., Line 123).

10、 Line 410: “…two rounds of centrifugation… were undergone in order to remove cells…” The passive voice “were undergone” is inappropriate and unnatural.

11、 Line 113: “CREB3L4. and its family member CREB3L2” contains an extraneous period after “CREB3L4.”

Author Response

Comments and Suggestions for Authors

1、The statistical analysis section (Lines 453–471) does not specify the software settings or assumptions (e.g., normality tests, multiple comparison corrections) used for the Mann-Whitney U-test and ROC curve analyses, which may affect reproducibility.

Answer: In the design of our analysis, we considered using non-parametric tests given that our sample size is moderately limited, and these tests do not assume a particular underlying distribution of data. In spite of this, we have additionally performed the Kolmogorov-Smirnov analysis to check if molecular data are normally distributed. We found that some of our variable data (PSA_blood, CREB3L4, CCNQ, PCA3) are not normally distributed, so that the use of non-parametric test for all variables was maintained as the best option to perform our analysis.

In our sEV-gene expression analysis, a limited number of genes (n=8) were included; the overall risk of Type I error remains acceptable and multiple comparison adjustment is not essential for an accurate analysis.

2、 Does the expression of long transcripts in semen sEVs vary due to other clinical factors in prostate cancer patients (e.g., age, treatment history, or prostate inflammation status)? It is recommended to include covariance analysis to rule out secondary confounding factors.

Answer: Selected patients had very similar clinical features: all of them are individuals in their diagnosis phase, thus, at the time of semen sample collection, there is still no treatment. Additionally, we have no evidence of prostatic inflammation in BPH or PCa patients.

Referring age variable, there is no difference in age between BPH and PCa patients (mean value 58.8 years old; Mann-Whitney p-value=0.906), thus age may not be a factor which influences the expression of long transcript analysis in semen EVs (Spearman correlation p>0.05) in those prostate-affected individuals. This is relevant for the clinics, when distinguishing PCa from benign prostate conditions such as BPH in those individuals with PSA levels between 4-10 ng/mL.

In order to obtain semen samples from healthy individuals and to fit with ethical issues, we collected samples from healthy individuals who consulted for vasectomy (inclusion criteria: healthy individuals older than 30 years of age that consulted for vasectomy -line 408 of the highlighted version of the manuscript-). Although selected HCt were mostly in their 40s, these individuals are still younger than BPH and PCa individuals and thus, differences in age were found between HCt and BPH_PCa individuals (Mann-Whitney p-value<0.0001). Considering all the samples in the study (HCt and BPH_PCa samples) a significant correlation between age and molecular variables was simultaneously found for citric acid (R Spearman: 0.309; p: 0.017), KLK3 (R Spearman: -0.412; p:0.001) and PCA3 (R Spearman: 0.556; p<0.0001), similarly as it occurs with PSA_blood (R Spearman:0.760; p<0.0001), although indicating a weaker correlation between age and sEV-transcript levels in semen than the one estimated for age and PSA-blood.

A paragraph was included in the manuscript for clarification (line 202-210 of the highlighted version of the manuscript).

3、 The manuscript interchangeably uses terms such as “semen sEVs,” “seminal sEVs,” and “sEV-contained transcripts” without clear definition or justification, which may cause confusion for readers.

Answer: “Semen and seminal sEVs” refer to extracellular vesicles in seminal fluid (a component of semen) with “seminal” referring to the fluid and “semen” being the overall sample containing both sperm and seminal fluid. We have included a phrase in the introduction (line 93 of the highlighted version of the manuscript) for clarification. The term “sEV-contained transcripts” refer to RNA molecules enclosed within extracellular vesicles.

4、 Lines 29–31: “Our findings also present strong evidence that classifiers based on combined long transcript levels in semen sEVs serve as effective biomarkers, alone or in combination with blood PSA and/or semen citric acid levels, for improving diagnosing PCa and assessing its severity and disease progression with high accuracy.” This sentence contains multiple clauses and concepts (classifiers, biomarkers, combinations, diagnosis, and prognosis), making it verbose

Answer: We thank the referee for pointing out this confusing paragraph. In this new version we have split it into two sentences to be better understood.

5、 Line 410: “…two rounds of centrifugation (1600x g for 10 min, then 16,000x g for 10 min at 4 ºC) were undergone in order to remove cells…” (past tense, but “were undergone” is inappropriate).

Answer: Sentence was modified for clarification

6、 Line 43: “…revealing patients with a suspicion of cancer.” The phrase “a suspicion” is inappropriate, as “suspicion” is typically an uncountable noun in this context.

Answer: We thank the referee for this grammatical clarification. We have omitted the article “a”: “…revealing patients with suspicion of cancer”

7、 Line 393: “…which prevented from a proper quantification.” The phrase “prevented from a proper quantification” lacks an article and is poorly worded.

Answer: We agree with the referee and have omitted this confusing phrase in this new version, as it is obvious that there is nothing to measure without a sample to analyse.

8、 Line 112: “After revising the list of their candidate biomarkers, we selected CREB3L4. and its family member CREB3L2…” The phrase “the list of their candidate biomarkers” can be simplified.

Answer: The sentence “After revising the list of their candidate biomarkers, we selected.…” has been replaced by “Among the candidate biomarkers they proposed, we selected…”

9、 Line 53: “4-10 ng/ml” (inconsistent with “ng/mL” used elsewhere, e.g., Line 123).

Answer: We have consistently changed ng/ml to ng/mL along the text

10、 Line 410: “…two rounds of centrifugation… were undergone in order to remove cells…” The passive voice “were undergone” is inappropriate and unnatural.

Answer: Sentence was modified for clarification

11、 Line 113: “CREB3L4. and its family member CREB3L2” contains an extraneous period after “CREB3L4.”

Answer: Sentence was modified accordingly.

Reviewer 2 Report

Comments and Suggestions for Authors

Although interesting, the study employed a small number of patients to investigate differences in the expression of proteins and non-coding RNAs in the seminal fluid. Prostate cancer and BPH are prevalent, and there is no difficulty in getting more patients to improve the results. Authors must demonstrate how this panel can replace the existing molecular tests for the same purpose as Decipher and Oncotype, which already have clinical acceptance and have been submitted for validation for years. 

Author Response

Comments and Suggestions for Authors

1. Although interesting, the study employed a small number of patients to investigate differences in the expression of proteins and non-coding RNAs in the seminal fluid. Prostate cancer and BPH are prevalent, and there is no difficulty in getting more patients to improve the results.

Answer: Fortunately, prostate cancer is a slow-growing tumour. As the chance of having prostate cancer for men with PSA levels between 4-10 ng/mL (the target of our study) is approximately 1 in 4, the recommendation for prostate biopsy can be postponed if there is no other symptom, abnormal DRE findings or clinically meaningful lesions on mpMRI (multi-parametric magnetic resonance imaging) analysis. The decision depends on a comprehensive clinical risk assessment. This decision, if postponed, can make difficult the selection of prostate biopsy candidates within the PSA range 4-10ng/mL to be enrolled in the study, given the time frame of our proposal.

2. Authors must demonstrate how this panel can replace the existing molecular tests for the same purpose as Decipher and Oncotype, which already have clinical acceptance and have been submitted for validation for years. 

Answer: Decipher and Oncotype Prostate tests are tissue-based genomic tests that are used to complement traditional clinical diagnostic features. Decipher results are based on the expression behaviour of 22 genes whereas Oncotype relies on the expression behaviour of 17 genes on biopsy/prostatectomy prostatic tissue.

Our proposal is based on non-invasive prostate cancer–specific biomarkers to complement PSA-related cancer screening and that can distinguish between the aggressive prostate cancer tumour type and the indolent prostate cancer form before the biopsy performance. In this way, overdiagnosis of PCa could be reduced and individuals requiring prostate biopsy confirmation could be more accurately selected.

This idea has also included in the discussion section of the current version of the manuscript (lines 350-359).

Reviewer 3 Report

Comments and Suggestions for Authors

Ferre-Giraldo et al.,  have studied semen extracellular vesicle markers which can help to improve the aggressiveness of prostate cancer. Overall, the manuscript flows very well. Here are a few clarifications that need further addressed before publication:

  • Fgure2, age , the normalized RNA expression levels for KLK3, CREB3L2 and PCA3 is very low as compared to CREB3L4, CCNQ, DUSP23, ACTG1, MXD4., etc. Does normalized value reflect Ct value of the qPCR. Authors should clarify. KLK3 expression HCt vs PCa has very narrow range difference 0.15 vs 0.20 . Is KLK3 can be ideal marker for clinical diagnostics.  Author should add raw Ct values along with internal control to interpret real significant differences between Healthy, BPH and PCa panels.
  • Figure 3, it is very hard to interpret results at transcript level of CREB3L4, CCNQ, KLK3 and PCA3 due to few outlier samples of prognostic group. Author should perform ELISA to observe basal protein difference for projected all markers in Figure 3 to conclude their findings.  
  • Is there specific rationale for normalizing qPCR transcripts with RPL19 and RPS17? Are they specifically expressed on semen fluid or extracellular vesicles? 2dCp strategy is like 2-ΔΔCT or different? Author should clarify in method section.

Author Response

Comments and Suggestions for Authors

Ferre-Giraldo et al., have studied semen extracellular vesicle markers which can help to improve the aggressiveness of prostate cancer. Overall, the manuscript flows very well. Here are a few clarifications that need further addressed before publication:

  • Fgure2, age , the normalized RNA expression levels for KLK3, CREB3L2 and PCA3 is very low as compared to CREB3L4, CCNQ, DUSP23, ACTG1, MXD4., etc. Does normalized value reflect Ct value of the qPCR. Authors should clarify. KLK3 expression HCt vs PCa has very narrow range difference 0.15 vs 0.20 . Is KLK3 can be ideal marker for clinical diagnostics.  Author should add raw Ct values along with internal control to interpret real significant differences between Healthy, BPH and PCa panels.

Answer: Contrary to the referee’s comment, the expression levels for KLK3, CREB3L2 and PCA3 are higher than the levels observed for CREB3L4, CCNQ, DUSP23, ACTG1, MXD4 (we apologize that the negative symbol is hardly appreciated in the figure; the scientific number is changed to general number in the current version).

In this version we include a new supplementary figure (Suppl. Figure 1 in the current highlighted version) showing the Cq/Ct values as the referee suggests.

  • Figure 3, it is very hard to interpret results at transcript level of CREB3L4, CCNQ, KLK3 and PCA3 due to few outlier samples of prognostic group. Author should perform ELISA to observe basal protein difference for projected all markers in Figure 3 to conclude their findings.  

Answer: We agree with the reviewer that there is hard to interpret Figure 3 individually due to the low number of samples of each prognostic group. From this analysis/figure we chose those genes that were dysregulated in some of the comparisons (Mann-Whitney test) when samples are classified by their prognosis. To determine their diagnostic performance of single mRNAs we determine their predictive accuracy to distinguish between low risk PCA (BPH+PCaI+PCAIIa+PCaIIB) vs high risk cancer [PCaIIC+PCaIIIB) (Table 2E) and propose the study of multivariate versus individual model.

As refers to ELISA study, the aim of the current work is not to perform a proteomic but a transcriptomic analysis of EVs in semen that can reflect the health of the cell of origin. It is important to take into consideration that these EV-derived transcripts can be taken up by recipient cells, (where they can be translated into proteins inducing different biological responses), contributing to cellular communication contributing to disease progression. Thus, transcript and protein levels in EVs may not be directly correlated.

Additionally, in the context of EVs, ELISA procedure is usually applied to characterize EVs by highly abundant proteins. Unfortunately, most of EV contained proteins are present at low concentration which presents a challenge for detection.

  • Is there specific rationale for normalizing qPCR transcripts with RPL19 and RPS17? Are they specifically expressed on semen fluid or extracellular vesicles? 2dCp strategy is like 2-ΔΔCT or different? Author should clarify in method section.

Answer: The reference genes were selected in a previous study because of their low interpatient variability and their robust analytical performance (Bonache et al., 2012. Doi: 10.1093/humrep/des074). We have additionally checked the expression behaviour of these genes in our semen EV sample cohort that resulted in corroborating that the mean of RPL19 and RPS17 expression values present a low coefficient of variation among samples and are among the most stable expression values (Supplementary Table S2 is now included in the manuscript).

Round 2

Reviewer 3 Report

Comments and Suggestions for Authors

Authors have addressed all the concerns raised during review process.